# Changes in Plasma Lipid Levels Following Cortical Spreading Depolarization in a Transgenic Mouse Model of Familial Hemiplegic Migraine

**DOI:** 10.3390/metabo12030220

**Published:** 2022-03-01

**Authors:** Inge C. M. Loonen, Isabelle Kohler, Mohan Ghorasaini, Martin Giera, Arn M. J. M. van den Maagdenberg, Oleg A. Mayboroda, Else A. Tolner

**Affiliations:** 1Department of Human Genetics, Leiden University Medical Center, 2333 ZC Leiden, The Netherlands; i.c.m.loonen@lumc.nl (I.C.M.L.); a.m.j.m.van_den_maagdenberg@lumc.nl (A.M.J.M.v.d.M.); 2Center for Proteomics and Metabolomics, Leiden University Medical Center, 2333 ZC Leiden, The Netherlands; i.kohler@vu.nl (I.K.); m.ghorasaini@lumc.nl (M.G.); m.a.giera@lumc.nl (M.G.); o.a.mayboroda@lumc.nl (O.A.M.); 3Department of Neurology, Leiden University Medical Center, 2333 ZC Leiden, The Netherlands

**Keywords:** cortical spreading depolarization, familial hemiplegic migraine, metabolomics, lipid mediators

## Abstract

Metabolite levels in peripheral body fluids can correlate with attack features in migraine patients, which underscores the potential of plasma metabolites as possible disease biomarkers. Migraine headache can be preceded by an aura that is caused by cortical spreading depolarization (CSD), a transient wave of neuroglial depolarization. We previously identified plasma amino acid changes after CSD in familial hemiplegic migraine type 1 (FHM1) mutant mice that exhibit increased neuronal excitability and various migraine-related features. Here, we aimed to uncover lipid metabolic pathways affected by CSD, guided by findings on the involvement of lipids in hemiplegic migraine pathophysiology. Using targeted lipidomic analysis, we studied plasma lipid metabolite levels at different time points after CSD in wild-type and FHM1 mutant mice. Following CSD, the most prominent plasma lipid change concerned a transient increase in PGD_2_, which lasted longer in mutant mice. In wild-type mice only, levels of anti-inflammatory lipid mediators DPAn-3, EPA, ALA, and DHA were elevated 24 h following CSD compared to Sham-treated animals. Given the role of PGs and neuroinflammation in migraine pathophysiology, our findings underscore the potential of monitoring peripheral changes in lipids to gain insight in central brain mechanisms.

## 1. Introduction

Migraine is a very common, highly disabling brain disorder, characterized by recurrent attacks of typically unilateral headache and associated neurological symptoms [1,2]. In about one-third of patients, the headache phase is preceded by transient focal neurological symptoms referred to as the aura [2]. The electrophysiological correlate of the aura is cortical spreading depolarization (CSD), a transient wave of intense neuronal and glial depolarization that slowly propagates through the cortex, which is followed by a long-lasting neuronal inactivity [3]. CSD events have been associated with the release of neuropeptides [3] and various brain metabolite changes, as shown for example by mass spectrometry (MS) imaging of brain sections of mice [4]. Due to CSD-related blood–brain barrier disruption [3,5,6], CSD-induced metabolite changes in the brain may be traceable in peripheral body fluids. In earlier work, we investigated CSD-related plasma metabolic changes using capillary electrophoresis—mass spectrometry (CE–MS) followed by multivariate data analysis, and observed changes in plasma levels of lysine and its by-product pipecolic acid in a mouse model of familial hemiplegic migraine type 1 (FHM1) [7]. FHM1 is a monogenic subtype of migraine with aura caused by specific missense mutations in the *CACNA1A* gene that encodes the pore-forming α_1A_ subunit of neuronal Ca_V_2.1 voltage-gated calcium channels [8]. FHM1 mutant knock-in mice carrying the human pathogenic R192Q missense mutation (‘R192Q mutant mice’) exhibit an enhanced Ca_V_2.1 channel function and consequently a higher level of glutamatergic neurotransmission in the cortex, resulting in an enhanced susceptibility to experimentally induced CSD [9,10,11]. In the FHM1 mutant mice, CSDs are associated with an increased neuroinflammatory response, either immediately following CSD [12] or lasting for at least one day after CSD [13]. 

To further explore whether peripheral metabolic changes can be detected in response to CSD, also in the context of rare hemiplegic migraine, we here targeted a specific class of regulatory lipids, namely pro-resolving lipid mediators [14], of which we measured plasma levels at different time points following CSD. This targeted approach was chosen based on previous findings on lipid mediators that were indicated to play an important role in migraine pathophysiology, being involved in vasodilatation [15] and neuro-inflammation [16], and which have been related to increased attack severity [17] and frequency ([18,19,20,21,22] for review). However, little is known about the mechanisms underlying these processes at the metabolome level. Profiling CSD- and FHM1-related changes in plasma lipid metabolites in a quantitative manner, which is only possible in a targeted metabolomics approach, may offer a clinically applicable approach to obtain insight into pathophysiological mechanisms and may identify plasma lipid mediators as putative disease biomarkers. In patients with common migraine, such an approach revealed a migraine-associated decrease in high-density lipoprotein (HDL) and omega-3 fatty acid plasma levels [23] and various changes in serum lipid metabolites [24]. During migraine attacks, elevated levels of prostaglandins PGE_2_, PGD_2__α_ and PGF_2_ have been reported in saliva of patients [25,26], and for PGE_2_ also in plasma [27]. Based on the indicated role of lipids in migraine pathophysiology, we here carried out a targeted analysis of pro-resolving lipid mediators in plasma following CSD induction in WT and FHM1 mutant mice. Using reversed-phase liquid chromatography-tandem mass spectrometry (RPLC-MS/MS), absolute concentrations of lipids in plasma were assessed at different time points after CSD induction, that is, for a short-term period up to 1.5 h and for a long-term period up to 48 h after CSD.

## 2. Results

### 2.1. CSD Characteristics and Physiological Parameters 

The study involved two independent sets of mouse experiments, covering short- and long-term time frames of blood sampling during and following induction of 7 CSD events in the cortex of mice during a 1 h period, by topical application of 300 mM KCl (or 300 mM NaCl in the Sham procedure) (Figure 1A,B). The 7-CSD induction paradigm was chosen mainly as it increases chances of capturing CSD-associated changes in peripheral lipid levels and allowed for blood sample collection in the immediate phase of CSD initiation (with a sample collected after the 4th CSD), directly after the CSD induction period (i.e., sample after the 7th CSD), and at multiple time points up to 2 days after CSD induction. In the short-term experiments, blood sample collection was performed up to 1.5 h following CSD, from anesthetized mice (as mice were kept under anesthesia after the CSD induction method until blood withdrawal), whereas in the long-term experiment, sample collection was performed up to 48 h following CSD, from awake mice (as mice needed to be woken up from anesthesia). CSD characteristics amplitude and half-width duration did not differ between WT and R192Q mutant mice for both the short-term and long-term experiments (Table 1). Physiological parameters pH, pCO_2_, pO_2_, mean arterial blood pressure that were assessed during the short-term experiments, were within physiological ranges and did not differ between genotypes and treatment groups (Table 1).

### 2.2. Targeted Quantitative Analysis of Lipid Mediators in Peripheral Plasma Following CSD

To decipher the contribution of lipid mediators to consequences of CSD, absolute plasma concentrations of adrenic acid (AdA), docosahexaenoic acid (DHA), arachidonic acid (AA), eicosapentaenoic acid (EPA), docosapentaenoic acid omega-3 (DPAn-3), linoleic acid (LA), linolenic acid (ALA), 10-hydroxydocosahexaenoic acid (10-HDHA), 7-HDHA, 17-HDHA, leukotriene E_4_ (LTE_4_), 5-hydroxyeicosatetraenoic acid (5-HETE), 8-HETE, 11-HETE, 12-HETE, 15-HETE, 14,15-diHETE, 19,20-dihydroxydocosapentaenoic acid (19,20-DiHDPA), and prostaglandin D2 (PGD_2_) were determined using a targeted RPLC-MS/MS approach. PGE_2_, 8(9)-epoxyeicosatrienoic acid (8(9)-EET), 11(12)-EET, 14(15)-EET, 9-hydroxyoctadecadienoic acid (9-HoDE), 13-HoDE, 9-hydroxyoctadecatrienoic acid (9-HoTrE) and 13-HoTrE showed a signal higher than the limit of detection (LOD) but lower than the validated lower limit of quantitation (LLOQ) of the quantitation method so instead of the concentration, the peak area was determined. Appendix A provides a list of all targeted (and measured) lipid mediators, structured by metabolic class.

Firstly, we evaluated the lipid concentrations in the plasma samples that were collected from untreated mice, so prior to the start of the CSD experiments. For some of the metabolites quantified, a relatively large inter-individual biological variability was observed between animals within each group. To allow for a comparison across metabolites and experimental groups, these metabolites data were reported as the ratio between the metabolic product and the substrate of a shared metabolic pathway, reflecting the enzymatic activity. Figure 2 shows a volcano plot built on the comparison of the plasma lipid mediators in WT and R192Q mutant mice that revealed no statistically significant differences in lipid concentrations between the two genotypes at baseline. Appendix A shows the abundance of the plasma lipids in a heatmap from the baseline samples. Accordingly, no genotype-specific difference was revealed. 

### 2.3. PGD_2_ Shows a Transient Increase Following CSD That Persists Longer in R192Q Mutant Mice

Considering the complexity of the experimental approach (two genotypes, two intervention groups and several time points), we addressed the time-specific changes in response to CSD induction within each genotype (WT and R192Q mutant) using a two-factor (time and intervention) repeated measures ANOVA test. Using F-factor and *p*-value of the ANOVA models as selection criteria, only PGD_2_ levels were found to change significantly over time following CSD, showing a transient increase in both genotypes (WT: F = 6.814, *p* = 0.007, RQ: F = 12.496, *p* = 0.001). Figure 3 shows the time course of the observed increase in PGD_2_ for WT (Figure 3A,B) and R192Q mutant mice (Figure 3B,C). In both WT and mutant mice, PGD_2_ levels were enhanced compared to levels in the Sham group at 0.5 h following the start of CSD induction, while for WT animals, PGD_2_ levels were comparable to levels in the Sham group at 1 h, plasma PGD_2_ concentration remained significantly higher at 1 h for R192Q mutant mice and had returned to the level of the Sham group only at 1.5 h. Within the intervention groups (CSD or Sham), no differences in PGD_2_ levels were detected at 0.5 h between WT and R192Q mutant mice (CSD–WT: median = 0.0532, interquartile range (IQR) = 0.023, CSD–RQ: median = 0.0585, IQR = 0.004, *p* = 0.403; Sham–WT: median = 0.0072, IQR = 0.025, Sham–RQ: median = 0.0357, IQR = 0.004, *p* = 0.21; indicated *p*-values obtained from a Mann–Whitney U-test). 

### 2.4. Long-Term CSD-Related Changes in the Plasma Lipid Metabolome 

For the long-term experiments with blood sampling performed from 4 to 48 h following the start of CSD induction, PGD_2_ plasma concentrations remained stable and did no longer show a CSD-related increase (Figure 4). With respect to other lipid mediators, four lipids showed a transient increase in WT mice at 24 h following CSD in comparison to the Sham group, i.e., DPAn-3 (F = 6.736, *p* = 0.007), EPA (F = 5.506, *p* = 0.0151), ALA (F = 4.672, *p* = 0.025) and DHA (F = 4.44, *p* = 0.029; Figure 5). This transient increase was not observed for R192Q mutant mice (see Appendix A for DPAn-3, EPA, ALA, and DHA concentrations at all time points for WT and R192Q mice). When comparing the 24 h plasma concentrations of these four lipids following CSD between genotypes, no significant differences were found (Figure 6).

## 3. Discussion 

Using a translational approach by investigating effects of CSD in WT and FHM1 R192Q mutant mice, we set out to uncover novel plasma lipid changes that could increase our understanding of migraine pathophysiology and serve as possible disease biomarkers, in particular with respect to hemiplegic migraine and the migraine aura phase. To this end, we used a targeted approach to assess changes in lipid concentrations in plasma samples over time (both short-term and long-term) up to 48 h after experimentally induced CSD, as a correlate of the migraine aura. Under naïve, untreated conditions, WT and FHM1 mutant mice did not show differences in plasma concentrations of the measured lipid mediators. However, following induction of multiple CSD events within a 1 h time period, in anesthetized mice, plasma levels of PGD_2_ were increased at 30 min after the first CSD in both WT and FHM1 mutant mice. Levels returned back to baseline within 1–1.5 h following the first CSD, which took longer in the mutant mice. In long-term experiments at 4 h, 24 h and 48 h after CSD, PGD_2_ plasma levels were no longer increased, whereas plasma concentrations of DPAn-3, ALA, DHA and EPA showed a slight and transient increase at 24 h after CSD induction in WT mice only. 

Lipids and lipid mediators are essential for normal functioning of many processes in the brain, such as structural development, energy metabolism, and mediation of synaptic plasticity [28]. In the context of disease, changes in brain lipid metabolism have been implicated in the pathophysiology of several brain disorders including migraine and comorbid disorders stroke and epilepsy ([22] for review). In the same mutant mouse model studied here, as it is also considered a model of ischemic stroke [29], the occurrence of spreading depolarizations surrounding the infarcted cortical area [30] were associated with changes in the lipid profile in brain tissue [31]. Moreover, for both mice [32] and patients following stroke [33] lipid changes were observed in plasma. In a rat model for temporal lobe epilepsy, lipid concentrations measured both in the hippocampus and plasma revealed upregulation of DHA [34]. These findings underscore the potential of plasma lipids as biomarkers for changes in brain activity. Such changes include excessive neuronal activity known to occur during seizures and around infarct areas, as well as changes in brain activity in the context of migraine during the aura phase, likely as the result of CSD [35].

Prostaglandins are lipid compounds that are localized in brain structures known to be relevant to migraine pathophysiology, including the cortex, thalamus, brainstem and trigeminal ganglia, as well as cerebral blood vessels [19]. The relationship between prostaglandins and migraine headache was first observed in 1968 when Carlson et al. [36] found that intravenous PGE_1_ injection resulted in migraine-like headaches in eight healthy subjects, of which one person reported a unilateral headache accompanied by visual symptoms resembling a migraine aura. Prostaglandins are generally believed to mediate migraine headache initiation based on their known role of inducing dilatation of cranial vessels [19]. In the present study, in our FHM1 mouse model, we found that PGD_2_ was upregulated in mice that underwent CSD. PGD_2_ is able to dilate the human middle cerebral artery [37,38], to induce dural and peripheral sensory afferent sensitization, albeit to a lesser extent than PGE_2_ [19,39], and to induce migraine-like headache [38]. Platelets of headache-free migraineurs were reported to have lower amounts of PGD_2_ when compared to healthy individuals [40]. This finding related to vasoconstriction of cerebral arterioles measured interictally in migraine patients without aura, which was even more pronounced in migraine patients with aura [41]. In primary cultures of rat trigeminal neurons, PGD_2_ was shown to induce neuronal release of CGRP [42], a neuropeptide released upon trigeminal nerve activation and linked to neuroinflammation and pain in migraine headaches [43]. In the context of CSD, the increased levels of PGD_2_ observed at 30 min after CSD initiation may reflect a brain neuroinflammatory response, which has been reported to occur after CSD [44], and can induce the release of inflammatory mediators including PGD_2_ [19]. If such process would occur following CSD during the aura phase in patients, this can potentially activate vessel dilatation and activate sensory afferents leading to a migraine headache. Our findings suggest that PGD_2_ could have the potential to serve as a biomarker for aura-related changes, when measured in patients in plasma following the migraine aura phase. Since we found this transient increase to last longer in the FHM1 mutant mice, a longer duration of elevated PGD_2_ levels following an aura in patients might be indicative of specific changes in the context of hemiplegic migraine. The lowered PGD_2_ levels measured interictally in migraine patients [40] seem to underscore the biomarker potential of PGD_2_ also for periods in between migraine attacks.

The contrast between increased plasma levels of PGD_2_ and unaltered levels of PGE_2_ following CSD in our experiments seems in line with clinical and preclinical data that underscore distinct mechanisms and roles of the different types of prostaglandins in the context of migraine [19]. While infusion of PGD_2_ has been shown to provoke similar migraine-headache symptoms in healthy volunteers, as does infusion of PGE_1,_ PGE_2_ or PGI_2_ [15,36,38,45], vasodilatory effects were reported to be more pronounced for PGD_2_ compared to other prostaglandins [19,38]. In patients such provocation experiments have only been performed using PGE_2_ and PGI_2_, and only for migraine patients without aura, whereby the majority of patients developed migraine-like attacks [46,47]. The change in PGD_2_ following CSD, in the present study, may suggest the involvement of PGD_2_ in CSD-induced dilatation of cranial vessels with relevance to migraine with aura. It will therefore be of interest in future clinical studies to compare infusion effects of the different prostaglandins in migraine patients with and without aura.

The absence, in our study, of a change in PGE_2_ levels as well as of other blood lipid metabolite changes reported in clinical studies [19,20,22,24] may reflect that only selected metabolites are altered following CSD, which can be of relevance for understanding differences in pathways underlying the headache phase in attacks of migraine with and without aura. The absence of a change in PGE_2_, as well as other prostaglandins besides PGD_2_, seems unexpected given their downstream position with respect to arachidonic acid (AA) [19], of which levels were shown to be transiently elevated for a period of minutes in the cortex during CSD [48]. However, it should be noted that migraine-related changes in lipid mediators may be brain region specific, which could result in no apparent changes of such metabolites in plasma. For PGE_2_, such regional changes were observed in rats following nitroglycerin administration as experimental trigger of headache pathways, whereby reduced PGE_2_ levels were observed in the hypothalamus while levels in the brainstem were enhanced [49]. Additional reasons for differences between the limited amount of lipid changes in the present work and those of earlier (pre)clinical reports may include biomarker differences between plasma and serum [50] or variation of lipid mediators across the estrus cycle (given the use of female mice in our study) as reported for plasma PGE_2_ in the context of menstrual migraine [51].

About 1 day after CSD induction, when samples were collected in awake mice, we found an increase in plasma levels of DPAn-3, DHA, ALA and EPA in WT mice that was absent in FHM1 mutant mice. DHA, DPA, and EPA are polyunsaturated fatty acids that originate from desaturation and elongation enzymes from ALA. EPA is converted into DPA, which is further converted into DHA, with both processes being reversible [52]. ALA, DHA, and EPA are linked to the unsaturated fatty acid pathway, as are AA and linoleic acid that were among the targeted metabolites. Follow-up studies covering more metabolites of this pathway would, however, be required to enable a meaningful interpretation on the observed changes with respect to the known pathway interactions. In the brain, EPA and DHA inhibit neuroinflammatory processes in several ways, that is, (1) inhibition of the production of neuroinflammatory proteins such as tumor necrosis factor alpha (TNF-α) and various interleukins in various cell types, (2) reduction in the activity of mitogen-activated protein kinases (MAPKs), and (3) inhibition of signaling pathways resulting in lower macrophage responses [53]. FHM1 mutant mice, compared to WT, show more widespread neuroinflammatory changes in the brain at 30 min following CSD [12], as well as a unique cortical neuroinflammatory gene expression profile at 24 h following CSD [13]. Our finding of increased plasma levels of anti-inflammatory lipid metabolites at 24 h in WT mice, but not in FHM1 mutant mice, suggests a reduced anti-inflammatory response in FHM1 mutants. The absence of such an elevation in anti-inflammatory mediators in mutant mice suggests a neuro-inflammatory condition in the FHM1 mouse model, as has been published in, e.g., Franceschini et al. [54], perhaps to counteract the enhanced neuronal excitability in the brain of mutant mice. All four anti-inflammatory mediators have previously been associated with migraine, although not hemiplegic migraine. One study reported abnormally low serum levels of ALA in the majority of migraine patients [55]. Another study in which migraine GWAS data were investigated with respect to genetically associated blood metabolic traits, DHA and DPA were identified as fatty acids that were genetically associated with a higher risk for migraine [56]. Of note, lower dietary intake of DHA and EPA are associated with a higher attack frequency in migraine patients [18]; vice versa, increasing dietary intake of EPA and DHA in chronic and episodic migraine patients resulted in a reduction in headache frequency and severity [57] and physical pain [58], underscoring the involvement of these lipids in migraine pathophysiology. Our findings suggest that DPAn-3, DHA, ALA, and EPA might have potential as biomarkers of an anti-inflammatory response (when levels are elevated), or a neuroinflammatory response (in case of an absence of elevated levels) in the context of an aura and hemiplegic migraine.

Our findings underscore the potential of modulating the synthesis or actions of prostaglandins and reducing neuroinflammatory responses in the treatment of hemiplegic migraine and perhaps migraine with aura. With CSD as the correlate of the migraine aura, increased PGD_2_ levels after CSD could suggest that in patients such a change may occur following the aura phase. The prolonged duration of the PGD_2_ elevation in FHM1 mutant mice, compared to WT, may suggest a specific association of PGD_2_ with migraine susceptibility, in particular in the context of hemiplegic migraine. Putative novel migraine treatments based on our finding (after validation) could include targeting actions of PGD_2_, e.g., by blocking the receptors involved [19]. Since prostaglandins are synthesized in various brain cells from arachidonic acid by activated COX following stimulation [59], another therapeutic approach could be to intervene with prostaglandin synthesis by inhibition of COX, using nonsteroidal anti-inflammatory drugs, NSAIDs. Our observation of CSD-induced elevated anti-inflammatory lipid mediators DPAn-3, EPA, ALA, and DHA in WT (that were not altered in FHM1 mutant mice), suggests a reduced anti-inflammatory response in the context of hemiplegic migraine and CSD. This is in line with earlier work in which we observed specific CSD-related neuroinflammatory changes in the brain of FHM1 mutant mice [12,13]. In the context of treatment regimes, our findings appear in line with the use of NSAIDs in acute migraine treatment [60]. Thus, our combined findings of a CSD-related increase in PGD_2_ that was prolonged in FHM1 mutant mice, and the absence of CSD-related rises in anti-inflammatory lipid mediators in FHM1 mutant mice underscore the potential of anti-inflammatory treatment strategies. Such treatments would be expected to inhibit an increase in PGD_2_, which could arise during the aura phase, and stimulate an anti-inflammatory response that may be impaired in patients with hemiplegic migraine. Upon validation of our findings, additional strategies aimed at specifically targeting the pathways involving PGD_2_ and DPAn-3, EPA, ALA, and DHA may provide a basis for novel, more selective, treatments.

The design chosen for this study has some limitations. First, it was not feasible to estimate the sample size beforehand. A power analysis for longitudinal designs is complex because within-subject correlation strongly affects the estimated sample sizes, and given the low number of metabolic studies on peripheral and central diseases, which hampers an accurate power analysis. Second, the use of anesthetic and analgesic agents may have reduced our chances of finding CSD-associated lipid mediator changes. Using isoflurane anesthesia in the experiments and carprofen administration for the long-term survival experiments may have influenced our results. Isoflurane anesthesia has been shown to have inhibitory effects on lipid metabolism e.g., [61,62,63] and carprofen administration inhibits COX-2 that is involved in prostaglandin synthesis [64]. The increase in PGD_2_ levels, however, was no longer evident at 1 h after the first CSD in WT mice and at 1.5 h in R192Q mutant mice in the short-term experiments where carprofen was not used. The absence of elevated PGD_2_ levels at 1.5 h and later time points in the long-term experiments is therefore in line with the normalization of PGD_2_ at 1.5 h seen in the short-term data, making it less likely that carprofen suppressed levels of PGD_2_ levels in the long-term experiments. The analgesic effects of carprofen in mice have been reported to be diminished by 6 h after administration [65]; therefore, the effects of the presence of carprofen would not be expected at the 24 and 48 h sample collection time points. Whether levels of other prostaglandins such as PGE_2_ may have been suppressed by carprofen in the long-term experiments cannot be excluded. The absence of PGE_2_ changes in the short-term experiments may suggest, however, that CSD does not cause a specific increase in levels of this metabolite. By comparison of the CSD-treated animals with Sham-treated controls, we reduced the possible influence of isoflurane or carprofen on the outcomes, allowing us to relate the observed metabolite changes specifically to the prior occurrence of CSD. Nevertheless, it remains possible that the use of these agents may have masked CSD-associated effects on lipid metabolism, which could explain the limited amount of CSD-related changes in our study. Future studies in which CSD is induced in unanesthetized awake animals could be performed to overcome this potential issue. This would be possible, for example, by (repeated) optogenetic stimulation, by which single or multiple CSDs can be induced in freely behaving mice [66], allowing blood sample collection at both short-term and longer-term time points, during and after CSD induction, without the need for anesthetic and analgesic agents. Such future studies can help validating our findings, may identify additional CSD-associated metabolic changes, and can also address whether a single CSD may already be sufficient to yield the identified metabolite changes.

### Conclusions

This targeted study investigated changes in plasma concentrations of lipid mediators over time following induction of multiple CSD events in transgenic FHM1 mutant mice, carrying the FHM1 R192Q mutation, and WT controls. Plasma samples collected after CSD induction were analyzed by RPLC-MS/MS and revealed CSD-associated increases in lipid concentrations. That is, of PGD_2_ directly following CSD in both genotypes and, in WT mice only, increased lipid concentrations of DPAn-3, ALA, DHA and EPA at 24 h after CSD. Taken together, our findings are consistent with previously identified migraine-relevant lipid mediators associated with headache pain mechanisms and associated neuroinflammatory changes, and underscore the potential that peripheral plasma lipid mediator changes can give insights to central migraine mechanisms, in particular in the context of the aura phase and hemiplegic migraine. Given the overlap in their pathophysiology, the observed changes in peripheral body fluids may be of relevance also for other central nervous system disorders that are comorbid with migraine, such as epilepsy and stroke [67].

## 4. Material and Methods

### 4.1. Experimental Design and Animal Experiments

The main goal of the study was to investigate the CSD-related changes in plasma metabolic profiles at multiple time points after CSD in female homozygous *Cacna1a* FHM1 R192Q knock-in mice (‘R192Q mutant mice’) [9] and WT littermates. Mice (age 2 to 4 months) were kept under standard housing conditions with a 12 h light/dark cycle with food and water available ad libitum. Animal experiments were approved by the Animal Experiment Ethics Committee of Leiden University Medical Center. The experimental design, as illustrated in Figure 1, consisted of two paradigms, i.e., short-term and long-term experiments. For both paradigms, mice were randomly divided into four groups (*n* = 5), i.e., CSD and Sham treatment groups for R192Q mutant and WT mice. 

### 4.2. CSD Induction and Recording in Anesthetized Mice

Mice were anesthetized with 4% isoflurane in pressurized air (0.5 L/min) for induction and 1.5% for maintenance. For short-term experiments, a catheter was inserted into the left femoral artery for blood sampling and continuous blood pressure monitoring. Arterial blood gases (pCO_2_ and pO_2_) and pH were measured at the start and end of the recordings. In both short-term and long-term experiments, multiple (i.e., 7) CSD events were evoked as described by Shyti et al. [7] and Eising et al. [13] to have a pronounced exposure of the brain to CSDs and increases chances of finding CSD-associated changes in peripheral blood plasma samples. Briefly, mice were mounted in a stereotaxic frame (David Koph, Tujunga, CA, USA) in a shielded Faraday cage on a homeothermic heating pad with continuous monitoring of core temperature (37.0 ± 0.5 °C). A midline incision over the head was made to expose the skull. Two cranial windows were prepared at the following coordinates from bregma: 3.5 mm posterior, 2 mm lateral for CSD induction in the visual cortex (V1) and 0.5 mm anterior, 2 mm lateral for CSD recording in the motor cortex (M1). CSD events were induced by applying a cotton ball soaked in 300 mM potassium chloride (KCl) solution over the dura that was removed after 30 s, followed by thorough washing with saline solution. This procedure was repeated 6 times with 10 min intervals. Sham experiments involved the same procedure using 300 mM sodium chloride (NaCl) solution instead of KCl. CSD events were recorded as DC-potential changes using a glass electrode filled with 150 mM NaCl. Signals were filtered with a DC-500 Hz low-pass filter, 30X amplified and digitized at 1000 Hz sampling rate (PowerLab 16/30; AD Instruments Inc., Colorado Springs, CO, USA). 

### 4.3. Collection and Preparation of Plasma Samples

Blood samples were collected before CSDs were induced, during the CSD induction period, and up to 48 h after CSD induction, to capture plasma lipid level changes across both immediate and delayed time periods following CSD. For short-term experiments, all blood samples except for the first time-point (t-1h) were collected via the femoral artery catheter. The samples at t-1h were collected via tail cuts (arterial blood). Samples (40 µL) were collected in heparinized blood collection capillaries (Microvette CB 300, Sarstedt, Nümbrecht, Germany) ~1 h before the 1st CSD event (t-1h), right after the 4th (t-0.5h), and 7th (t-1h) CSD event, and 1.5 h after the 1st CSD (t-1.5h), and for Sham experiments at the same time points before or after the first NaCl application. For long-term experiments, all samples were collected via tail cuts, ~1 h before the 1st CSD event (t-1h), and 1.5 h (t-1.5h), 4 h (t-4h), 24 h (t-24h), and 48 h (t-48h) after the 1st CSD event, and at the same time points before or after the first NaCl application. Blood samples were centrifuged for 10 min with 4000 rpm at 4 °C prior to plasma collection. Plasma was transferred to cryotubes (Sarstedt), snap-frozen in liquid nitrogen, and stored at −80°C. In short-term experiments, mice were under isoflurane anesthesia for the entire experiment, for a period of ~2 h, and sacrificed directly after collection of the last sample. In long-term experiments, the mice were under isoflurane anesthesia for ~1.5 h for the surgery and CSD induction part of the experiment. In the long-term survival experiments, mice received a subcutaneous injection of carprofen (5 mg/kg), a non-steroidal anti-inflammatory drug to minimize the animal discomfort after surgery, directly after the 7th CSD while still under anesthesia. The skin overlaying the skull was sutured and 15 min after the carprofen injection, mice were allowed to wake up in a recovery cage equipped with a heating light. After the last sample collection (i.e., t-48h), mice were sacrificed.

### 4.4. Targeted Analysis of Lipid Mediators 

The targeted analysis of lipid mediators was performed using reversed-phase liquid chromatography-tandem mass spectrometry (RPLC-MS/MS) [14]. Samples were randomized prior to the analysis. Protein precipitation was performed using 25 µL MeOH added to 5 µL of plasma. The mixture was agitated and stored at −20 °C for 20 min prior to centrifugation at 16,000 rcf for 10 min at 4 °C. Some 25 µL of supernatant was collected and evaporated to dryness under a gentle steam of N_2_. Samples were reconstituted with 20 µL MeOH, agitated, prior to the addition of 30 µL of water. After agitation, 45 µL of the reconstituted samples were injected into the RPLC-MS/MS system. RPLC experiments were carried out using a Nexera LC system (Shimadzu, ‘s-Hertogenbosch, The Netherlands) equipped with a solvent degasser, a column oven, an autosampler with thermostat, and a quaternary pump. Separation was performed on a Kinetex C18 column from Phenomenex of 50 mm × 2.1 mm i.d., and 1.7 µm particle size. The mobile phase was composed of 0.01% acetic acid in water (A) and 0.01% acetic acid in MeOH (B). The flow rate was set to 400 µL/min with the following gradient profile: 30% B for 1 min, 30–45% B in 0.1 min, 45–53.5% B in 1.9 min, 53.5–55.5% B in 2 min, 55.5–90% B in 3 min, 90–100% B in 0.1 min, and 100% B for 2 min for a total analysis time of 9 min. Samples were kept at 6 °C and the separation was performed at 50 °C. A C_8_ Security Guard cartridge from Phenomenex of 20 mm × 2.00 mm i.d. was used as a pre-column. All the solvents used for the experiments were of LC-MS grade or higher. 

The RPLC system was hyphenated to a QTRAP 6500 LC-MS/MS system (SCIEX, Nieuwerkerk aan den IJssel, The Netherlands) equipped with a Turbo Spray IonDrive ESI source operating in negative mode. ESI capillary voltage was set at –4500 V. Nebulizing gas and curtain gas pressures were set at 40 psi and 30 psi, respectively. Heater gas temperature and pressure were fixed at 450 °C and 30 psi, respectively. Data was acquired in selected reaction monitoring (SRM) mode with a time window of ~30 s around the expected retention time. Dwell time and inter-channel delay were 150 ms and 5 ms, respectively. For all compounds, an entrance potential of −10 V was used. 

Deuterated internal standards (IS), i.e., 15-HETE-d_8_, DHA-d_5_, LTB_4_-d_4_, and PGE_2_-d_4_ were added prior to the protein precipitation to each sample at a final concentration of 1 ng/mL. All compounds were quantified using a calibration curve prepared in MeOH/water (40:60, *v*/*v*) consisting of 7 calibrants concentrations, each injected twice. Concentrations were estimated using the ratio of the peak area for each compound over the peak area of its respective IS. A weighing factor of 1/x and linear regression were used for quantitation. 

### 4.5. Statistical Analysis

Data were acquired using Analyst^®^1.6.2 build 8489 software and analyzed with MultiQuant^®^ version 2.1, and GraphPad Prism version 6.05. The R environment (R version 4.1.1) was used for data analysis and visualization, with the following software packages: tidyverse (1.3.1), broom.mixed (0.2.7), ggplot2 (3.3.5), ggsignif (0.6.3), superheat (0.1.0), and EnhancedVolcano (1.10.0). In comparing CSD features (amplitude and duration) in short-term and long-term experiments and physiological parameters across all experimental groups in the short-term experiment, one-way ANOVA was used for normally distributed datasets and the Kruskal–Wallis test for non-normally distributed datasets. Two factor (time and intervention) repeated measures ANOVA including correction for multiple testing was used to compare plasma lipid concentrations between genotypes over time in the RPLC-MS/MS experiments. The volcano plot, comparing baseline plasma lipid concentrations between genotypes (Figure 2), shows datapoints uncorrected for multiple testing. Subsequent comparison for selected metabolites at specific time points between CSD and Sham groups was performed with an unpaired two-tailed Mann–Whitney U test, since a minority of lipid metabolite datasets was not normally distributed. Data are expressed as individual measured values (concentration, area, concentration ratio, or area ratio depending on the fatty acid and enzymatic pathway) and median (horizontal line). 

## Figures and Tables

**Figure 1 metabolites-12-00220-f001:**
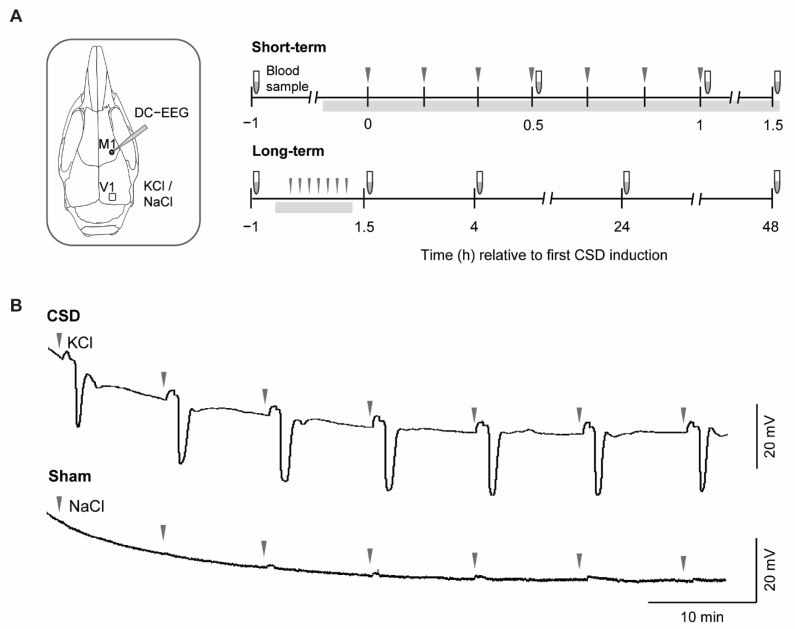
CSD experiments with short- and long-term plasma sample collection in mice, followed by plasma lipid metabolomics analysis. (**A**). Inset shows locations of topical application of KCl/NaCl over the visual cortex (V1) and the DC-EEG recording site in the motor cortex (M1). Topical application of 300 
mM KCl (for CSD induction) or NaCl (Sham procedure), as indicated by 
arrowheads, was performed every 10 min over a period of 1 h in 
isoflurane-anesthetized mice. Note that each application of KCl was sufficient 
to induce a single CSD event, thus in total 7 CSD events were induced within a 
1 h time window. In the schematic of the short-term and long-term experiments, 
the period during which mice were kept under isoflurane anesthesia is indicated 
by the grey bar underneath the timeline. In short-term experiments, mice were 
under isoflurane anesthesia for the entire experiment with a duration of ~2 h, 
and sacrificed directly after collection of the last sample. In long-term 
experiments, the mice were under isoflurane anesthesia for ~1.5 h during the surgery 
and CSD induction part of the experiment. For both short- and long-term 
experiments, baseline blood samples were obtained 1 h prior to CSD through a tail 
cut. In the short-term experiments, additional sampling was performed every 0.5 
h up to 1.5 h after the first CSD event by sampling from the femoral artery 
catheter in anesthetized mice. In the long-term experiments, sampling was 
performed at 1.5 h, 4 h, 24 h and 48 h after the first CSD, using tail blood 
samples from awake mice. (**B**) DC-EEG example traces of the CSD and Sham 
paradigm showing that CSD was induced only following KCl application and not 
after NaCl application (KCl or NaCl application moments indicated by 
arrowheads).

**Figure 2 metabolites-12-00220-f002:**
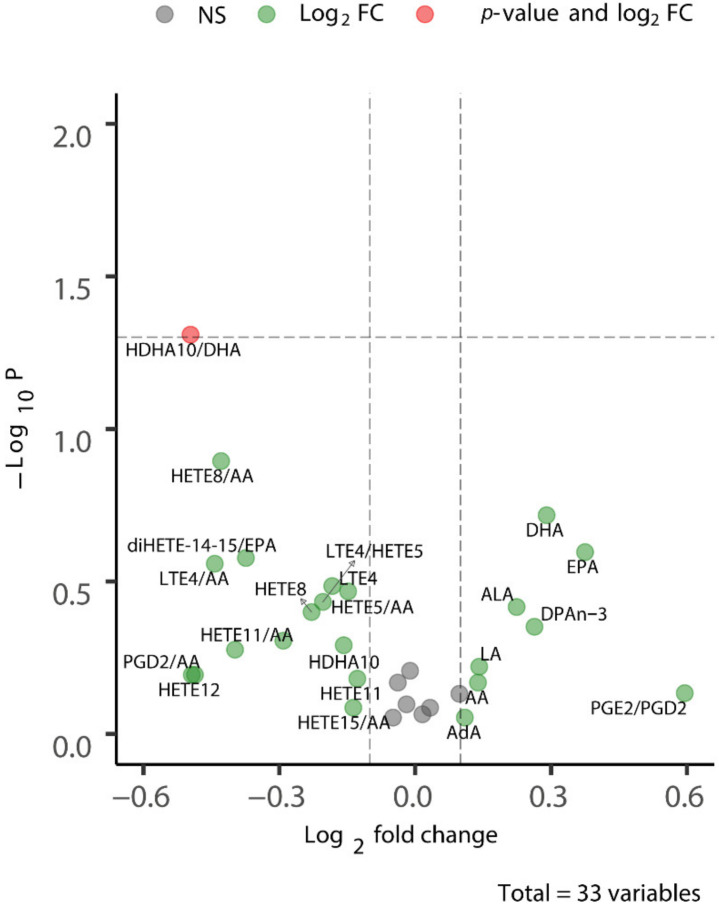
Baseline 
plasma lipid concentrations are similar between WT and R192Q mutant mice. 
The volcano plot shows that there are no significant differences in the 
concentration of measured plasma lipids between WT and R192Q mutant mice. For 
some metabolites, the ratio between product/substrate of a metabolic pathway 
was used (e.g., HDHA10/DHA, HETEB/AA, diHETE-14-15/EPA in the plot), given the large 
inter-individual variation in the absolute concentrations. See Appendix A for the list of targeted metabolites, 
structured by metabolic class; note that since the data analysis was conducted in 
the R environment, the metabolite names had to be adapted to make them readable 
for the software.

**Figure 3 metabolites-12-00220-f003:**
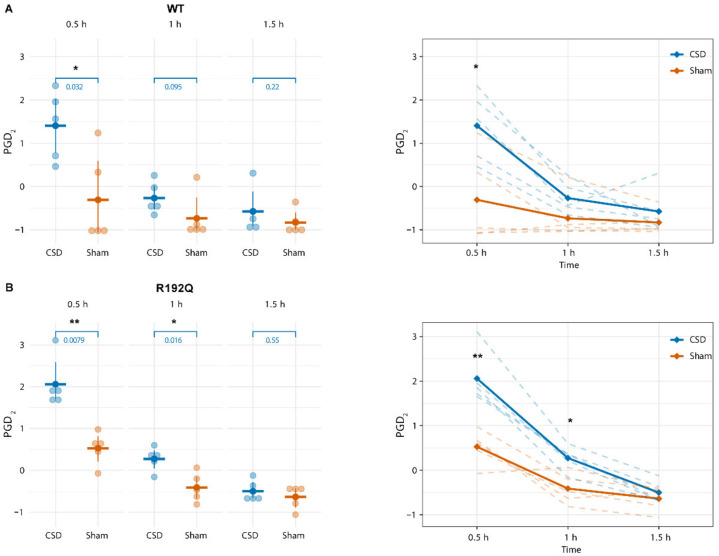
CSD is associated with a short-term transient increase in plasma PGD_2_ level. In both WT (**A**) and R192Q mutant mice (**B**), a transient increase in PGD_2_ was observed at 30 min following the first CSD induction compared to Sham-treated mice, that at 1 h was no longer evident in WT mice while remaining increased in R192Q mutant mice. For both WT and R192Q mutant mice, at 1.5 h following the first CSD, PGD_2_ levels were comparable to that of Sham-treated mice. Indicated *p*-values obtained from a Mann–Whitney U-test. Panels on the right show the time course of the levels of plasma PGD_2_ for WT and R192Q mutant mice for CSD compared to Sham groups. *: *p* ≤ 0.05; **: *p* ≤ 0.01.

**Figure 4 metabolites-12-00220-f004:**
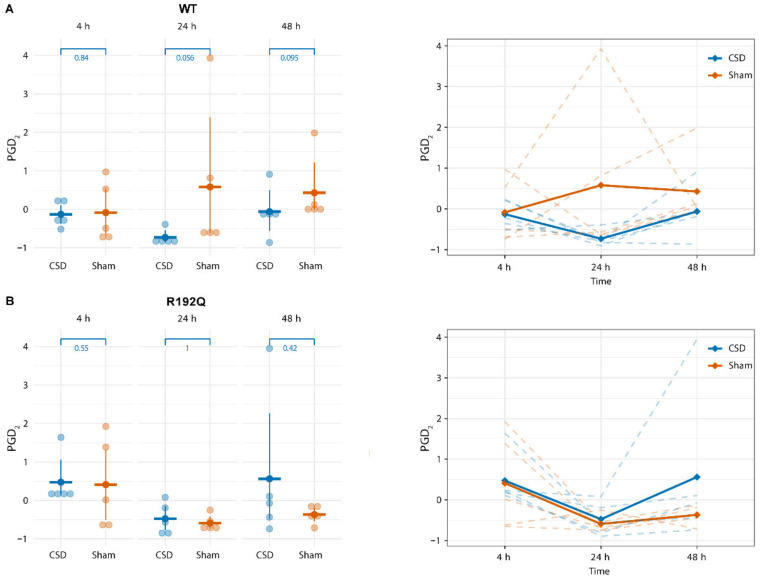
PGD_2_ plasma levels are not altered at long-term time points up to 48 h following CSD. For both WT (**A**) and R192Q mutant mice (**B**), there were no changes in PGD_2_ levels at 4 h, 24 h and 48 h following CSD, compared to levels of Sham-treated mice. Panels on the right show the time course of the levels of plasma PGD_2_ for WT and R192Q mutant mice for CSD compared to Sham groups.

**Figure 5 metabolites-12-00220-f005:**
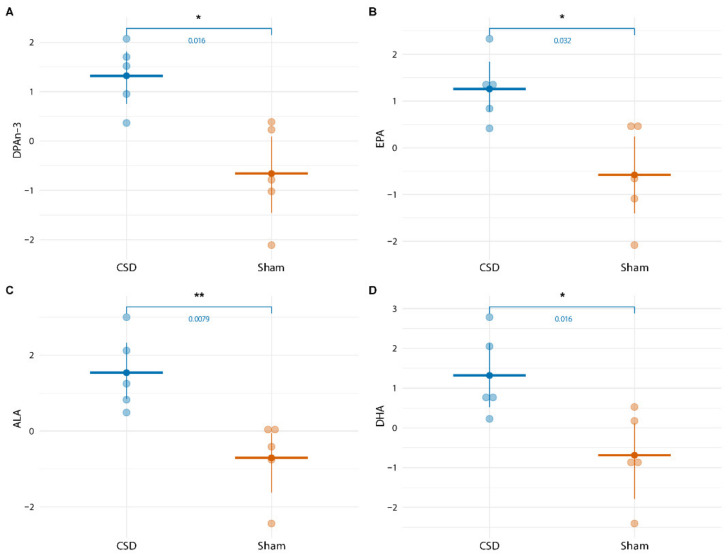
CSD-related increase in lipid concentration of DPAn-3 (**A**), EPA (**B**), ALA (**C**), and DHA (**D**) in WT mice at 24 h following CSD. In WT mice, levels of DPAn-3, EPA, ALA, and DHA were significantly higher at 24 h following CSD in comparison to levels in Sham-treated mice. *: *p* ≤ 0.05; **: *p* ≤ 0.01.

**Figure 6 metabolites-12-00220-f006:**
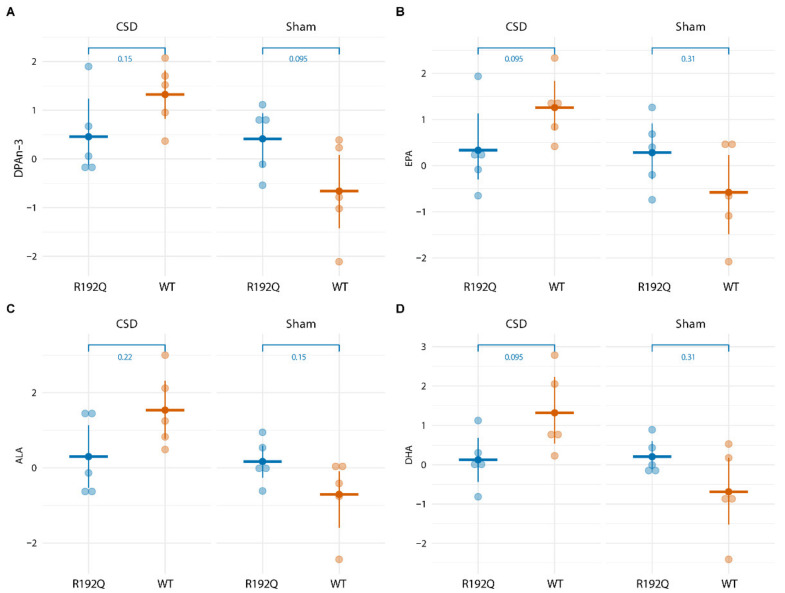
Plasma lipid concentrations of DPAn-3 (**A**), EPA (**B**), ALA (**C**), and DHA (**D**) compared between WT and R192Q mutant mice at 24 h following CSD did not differ.

**Table 1 metabolites-12-00220-t001:** CSD characteristics and systemic physiological parameters in catheterized mice during short-term experiments.

Experimental Groups	*n*	Amplitude (mV)	Duration (s)	pH	pCO_2_	pO_2_	MABP (mm Hg)
Short-term WT Sham WT CSD R192Q Sham R192Q CSD	5 5 5 5	- 18 ± 5 - 14 ± 6	- 47 ± 4 - 43 ± 6	7.34 ± 0.06 7.35 ± 0.02 7.36 ± 0.04 7.34 ± 0.01	32 ± 7 32 ± 5 28 ± 4 33 ± 4	115 ± 11 110 ± 12 120 ± 7 115 ± 9	77 ± 7 72 ± 4 74 ± 6 79 ± 7
Long-term							
WT Sham WT CSD R192Q Sham R192Q CSD	5 5 5 5	- 21 ± 3 - 22 ± 1	- 59 ± 19 - 54 ± 17	- - - -	- - - -	- - - -	- - - -

Values are shown as means ± SD. All physiological parameters during CSD recordings were within physiological ranges. For both CSD features and physiological parameters, group differences are not statistically significant (1-way ANOVA). Mean arterial blood pressure (MABP) and arterial partial pressure of carbon dioxide (pCO_2_) and oxygen (pO_2_) are expressed in mmHg, averaged over the recording period. *n*: number of mice; -: indicates that CSD features were not obtained during Sham experiments (since no CSD events were induced upon topical NaCl application), and physiological parameters were not obtained for long-term experiments (since no femoral artery catheterization was performed in these experiments).

## Data Availability

The data presented in this study are available in the article and Appendix A. Further information can be made available upon reasonable request to the authors.

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
