# Peer review of "Changes in Plasma Lipid Levels Following Cortical Spreading Depolarization in a Transgenic Mouse Model of Familial Hemiplegic Migraine"

_metabolites, 2022, doi:10.3390/metabo12030220_

Round 1
Reviewer 1 Report
Thank you for addressing the raised concerns. Some editing is still required, but the manuscript has improved.Author Response
Please see the attachment.

Reviewer 2 Report
The aim of this study was to examine the effect of cortical spreading depolarization (CSD) on plasma lipids in in familial hemiplegic migraine type 1 (FHM1) mutant mice using targeted lipidomic analysis. The results show that CSD characteristics amplitude and half-width 97 duration did not differ between WT and R192Q mutant mice. pH, pCO2, pO2, mean arterial blood pressure did not differ between genotypes and treatment groups. The results show that CSD transiently increase plasma prostaglandin D2 and this effect is more prominent in mutant than in wild-type mice. In addition, in the long run, CSD increased DPAn3, EPA, 31 ALA, and DHA but only in wild-type but not in mutant mice.
The topic and the results are of interest and the manuscript is well-written. I have only two comments how the manuscript could be improved.
- Statistical analysis: was normality of data distribution verified to choose between parametric and non-parametric tests?
- Title of section 2.4: it is stated in the title that PGD2 transiently decreased after CSD whereas the opposite is stated in the text and in the Abstract. This should be clarified.
- What clinical implications of the findings are proposed? Do the authors suggest to use some of these mediators as the markers in patients with migraine?
Reviewer 3 Report
The revised version has included clarification on several points. The authors have also discussed findings in a fair way and noted the study limitations in a balanced manner. There are no further comments.

Reviewer 4 Report
good interesting work
Author Response
Please see the attachment.

This manuscript is a resubmission of an earlier submission. The following is a list of the peer review reports and author responses from that submission.
Round 1
Reviewer 1 Report
Very interesting study with a lot of effort invested in the demonstration of the role of specific lipids in inducted headache.
However, I consider some specific points have to be clarified:
- Why did you choose to apply exactly 7 consecutive stimuli? One or 2 stimuli were not considered enough to generate a CSD?
- Please mention the total duration of anesthesia during the CSD induction and comment on the possible influence of isoflurane on lipid levels. Previous studies suggest a widespread and immediate effect of isoflurane on the glucose and lipid metabolism (including peripheral insulin resistance, decreased lipolysis, progressive protein wasting with increased duration of anesthesia).
- You chose to administer carprofen for minimizing discomfort after surgery. Please comment on the possible influence of carprofen on lipid levels and especially on PGD2 and PGE2 levels.
- Please elaborate on the possible application of your findings in the treatment (acute or preventive) of migraine.
Reviewer 2 Report
This is a useful targeted metabolomics study.
My only suggestion is to refrain generalizing the results' application to all types of migraine. This study was focused only on FHM mutant mice models, hence the interpretations and speculations only apply to FHM.
I was hoping to see some pathway analysis and relationships to FHM phenotypes, as metabolomics is the terminal downstream product before phenotype.
Why was targeted metabolomics selected instead of untargeted metabolomics? Are the identified metabolomics and targets applicable in humans as well? Not just mice metabolites?
Sample size estimation is missing. Must be mentioned as limitation.
Correction to multiple testing must also be included.